# Split selectable markers

Nathaniel Jillette[1,4], Menghan Du[1,2,4], Jacqueline Jufen Zhu[1], Peter Cardoz[1] & Albert Wu Cheng [ID] [1,2,3]*

Selectable markers are widely used in transgenesis and genome editing for selecting engineered cells with a desired genotype but the variety of markers is limited. Here we present split selectable markers that each allow for selection of multiple "unlinked" transgenes in the context of lentivirus-mediated transgenesis as well as CRISPR-Cas-mediated knock-ins. Split marker gene segments fused to protein splicing elements called "inteins" can be separately co-segregated with different transgenic vectors, and rejoin via protein trans-splicing to reconstitute a full-length marker protein in host cells receiving all intended vectors. Using a lentiviral system, we create and validate 2-split Hygromycin, Puromycin, Neomycin and Blasticidin resistance genes as well as mScarlet fluorescent proteins. By combining split points, we create 3- and 6-split Hygromycin resistance genes, demonstrating that higher-degree split markers can be generated by a "chaining" design. We adapt the split marker system for selecting biallelically engineered cells after CRISPR gene editing. Future engineering of split markers may allow selection of a higher number of genetic modifications in target cells.

[1] The Jackson Laboratory for Genomic Medicine, Farmington, CT 06032, USA. [2] Department of Genetics and Genome Sciences, University of Connecticut Health Center, Farmington, CT 06030, USA. [3] Institute for Systems Genomics, University of Connecticut Health Center, Farmington, CT 06030, USA. [4]These authors contributed equally: Nathaniel Jillette, Menghan Du. *email: albert.cheng@jax.org

Selectable markers, such as antibiotic resistance or fluorescent protein genes, are often used in genetic engineering to isolate cells with desired genotypes[1]. However, there are a limited number of well-characterized antibiotic resistance genes for use in eukaryotic cells and fluorescent proteins whose spectra can be unambiguously differentiated by commonly used equipment is similarly limited. Researchers often run into the problem of not having enough choices of selectable markers if they wish to incorporate multiple transgenes into a cell. On the other hand, selection with multiple antibiotics at the same time is often harsh to cells. "Selectable marker recycling" can provide a work-around but is unwieldy, requiring multiple rounds of transgenesis, selection and removal of markers[2].

To allow multiple transgene selection with a single scheme, we create here split antibiotics resistance and fluorescent protein genes. In this system, a gene encoding an antibiotic resistance or fluorescent protein is split into two or more segments and fused to inteins ("markertrons") that can be rejoined by protein trans-splicing[3] (Fig. 1). Each markertron is inserted onto a transgenic vector carrying a specific transgene. Delivery of transgenic vectors containing a set of markertrons yields cells that harbor either a subset or a complete set of the markertrons. Only cells with a complete set of markertrons produce a fully reconstituted marker protein via protein splicing and thus passes through selection while cells with partial sets of markertrons are eliminated, achieving co-selection of cells containing all intended transgenes.

## Results

**Intein-split antibiotic resistance (Intres) genes.** We began by engineering 2-markertron intein-split resistance (Intres) genes for double transgenesis. Since flanking residues and local protein folding can affect efficiency of intein-mediated trans-splicing, we set out to identify split points in each of the four commonly used antibiotic resistance genes compatible with two well-characterized split inteins derived from $NpuDnaE$[4,5] and $SspDnaB$[6]. To

facilitate assessment of the effectiveness of double transgenic selection, we cloned markertrons onto lentiviral vectors expressing TagBFP or mCherry fluorescent proteins as test transgenes (Fig. 2a). Viral preparations were transduced into U2OS cells, which were then split into replicate plates with non-selective or selective media. Following appropriate passages for antibiotics selection, the two cell cultures were analyzed by flow cytometry. For Hygromycin resistance (Hygro$^R$) gene, one "native" $SspDnaB$ split point (SspDnaB-200 = G200:S201; Plasmid pair 5,6) with flanking residues "GS" and one "native" $NpuDnaE$ split point ($NpuDnaE$-89 = Y89:C90; Plasmid pair 3, 4) with "YC" residues were tested (Supplementary Fig. 1a). Both enabled successful selection when both N- and C-markertrons were transduced yielding >95% BFP + mCherry + double transgenic cells in selected cultures compared to <40% double-positive cells in non-selected culture (Fig. 2b; Plasmid pairs 3, 4 and 5, 6). Cells transduced with either of the single markertrons did not survive Hygromycin selection. In contrast, double transgenesis with conventional full-length non-split Hygro$^R$ vectors only allowed for ~20% enrichment of BFP + mCherry + cells (Plasmid pairs 97,98) at lower titers and for up to ~50% at higher titers. We screened three additional potential split points (NpuDnaE-52 = 52 S:53 C; Plasmid pair 7,8), (NpuDnaE-240 = 240 A:241 C; Plasmid pair 9,10), and (NpuDnaE-292 = 292 R:293 C; Plasmid pair 11,12) for $NpuDnaE$ with the obligatory cysteine residue on the C-extein junction and a residue on the N-extein junction reported to support substantial trans-splicing activities[7]. We also tested six additional $NpuDnaE$ split points (NpuDnaE-69, 131, 171, 218, 259, and 277) by inserting an "artificial" cysteine on the C-extein junction to support splicing at ectopic sites yielding additional split points. In total, eight out of eleven split points tested supported Hygromycin selection (Fig. 2b). Two of the Hygro Intres designs (NpuDnaE-131 and 292) failed to provide resistance in two of the four replicate experiments at lower titers, while three designs (NpuDnaE-218, 259, and 277) failed to provide resistance in any experiments. These positions may reside within less efficient splicing sequence and structural contexts or may disrupt folding of the Hygro$^R$ protein upon reconstitution. Indeed, western blot analysis using terminally tagged markertron fragments revealed that among split points 52, 68, 89, 131, and 171, trans-splicing is least active at split point 131 (Supplementary Fig. 1b, lane 5). This is consistent with its failure to consistently provide resistance at a lower titer (Fig. 2b). In addition, the insertion of the artificial cysteine at the NpuDnaE-69, 131, and 171 C-markertrons is required for protein splicing mediated by $NpuDnaE$ intein at these positions (Compare lanes 2/3, 5/6, and 7/8), consistent with a well-established requirement[7]. Nonetheless, the six successful designs validate our screening strategy and demonstrate that Hygro$^R$ is amenable to splitting at different positions spanning a large portion of the protein. Similarly, for Puromycin resistance (Puro$^R$) (Fig. 3a), Neomycin/G418 resistance (Neo$^R$) (Fig. 3b) or Blasticidin resistance (Blast$^R$) (Fig. 3c) genes, we identified four, two, and one functional 2-split Intres pair(s), respectively. In all of these cases, cells transduced with single markertrons did not survive selection, while cells transduced with both yielded 88–100% double transgenic cells in selective cultures compared to <50% in non-selective cultures. Details of the split points of Intres genes and plasmids are presented in Supplementary Figs. 1–4 and Supplementary Table 1. To facilitate adoption of Intres markers, we created Gateway-compatible lentiviral vectors for convenient restriction-ligation-independent LR clonase recombination of transgenes[8] (Supplementary Fig. 5). We tested the functionality of these vectors by recombining TagBFP and mCherry, respectively to the N- and C-Intres vectors and found robust selection of double transgenic cells (Supplementary Fig. 5b). One potential utility of Intres

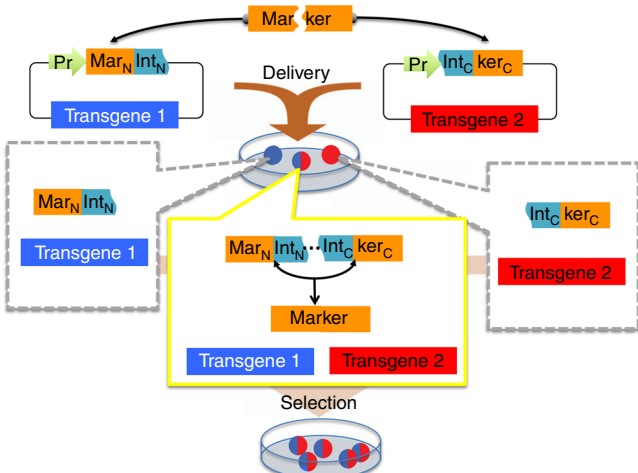

**Fig. 1** Split selectable marker for co-selection of two separate transgenic vectors. The coding sequence of a selectable marker is split into an N-terminal fragment (Mar$_N$) and a C-terminal fragment (ker$_C$) and separately cloned upstream of an N-terminal fragment of a split intein (Int$_N$) and downstream of a C-terminal fragment of the split intein (Int$_C$), respectively, on two different vectors each carrying a different transgene. These vectors are delivered to cells yielding sub-populations of cells containing either one, or both of the vectors. Only cells with both vectors expressing the two intein-split selectable marker fragments ("markertrons") undergo protein trans-splicing to reconstitute a full-length selectable marker, allowing specific selection and enrichment of the double transgenic cells

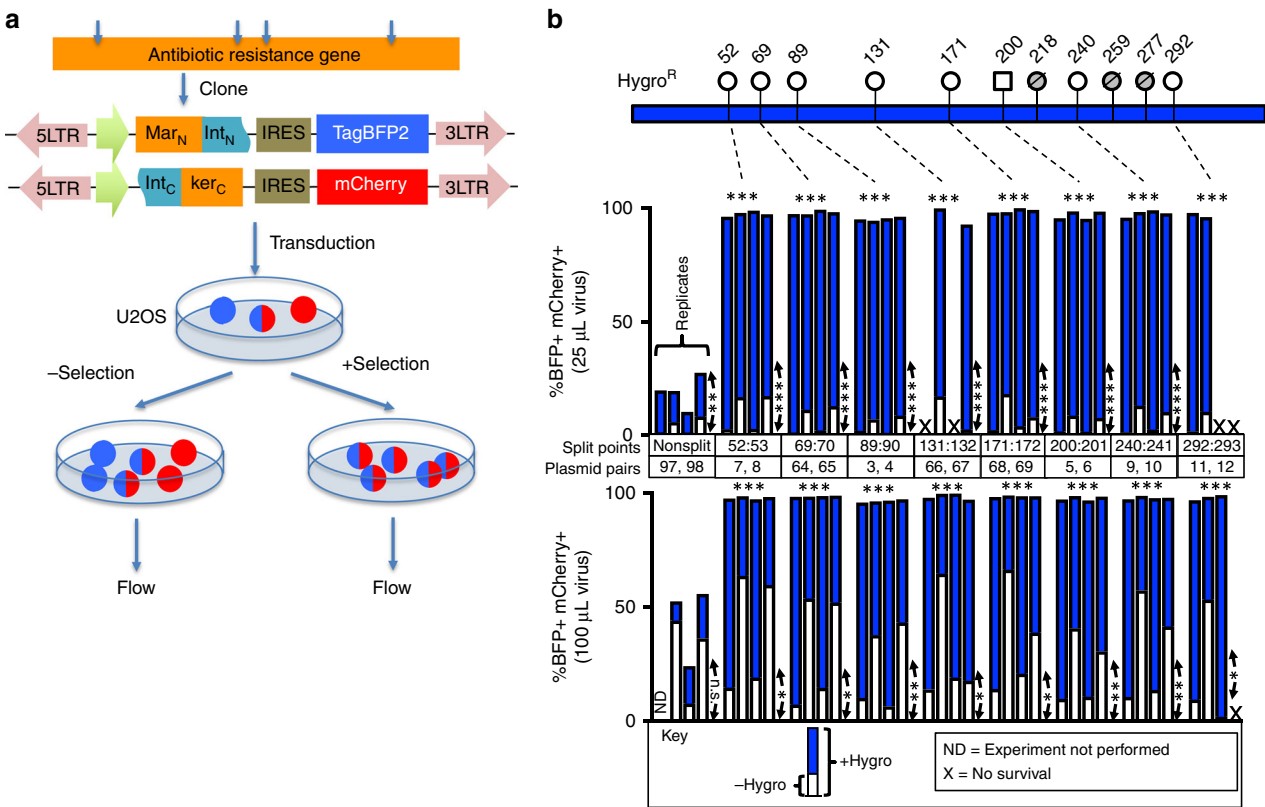

**Fig. 2** Split Hygromycin resistance genes for co-selection of two separate transgenic vectors. **a** To screen for split points compatible with inteins for an antibiotic resistance gene, we identified potential split points according to the junctional requirement for the type of intein tested, then cloned the corresponding N- and C-terminal fragments to the split intein scaffolds on lentiviral vectors equipped with TagBFP or mCherry fluorescent proteins, which serve as our test transgenes to evaluate selection efficiency. These are delivered into cells via lentiviral transduction. The cells were then split into replicate plates, one subjected to antibiotic selection while the other was maintained in non-selective media. Following antibiotic selection, the replicate cultures were analyzed by flow cytometry. **b** 2-split Hygromycin (Hygro) intein-split resistance (Intres) genes. Top schematics shows the split points tested for Hygromycin resistance gene. The last residue of the N-terminal fragment is indicated on top of the lollipops. Circle lollipops represent split points using *Npu*DnaE intein while square lollipops represent those using *Ssp*DnaB intein. Crossed-out and shaded lollipops indicate split pairs that failed to endow cells with Hygromycin resistance. The column plots below show the percentages of double transgenic cells (BFP+ mCherry+) in the non-selective (white portion) and the selective cultures (total column height = white + blue) quantified by flow cytometry, from samples transduced with 25 μL virus (middle column plot) or 100 μL virus (bottom column plot). Experiments were conducted in quadruplicate, with each column representing a completely independent virus preparation, transduction and selection. The 25 μL and 100 μL data in the same x-position represent the sister cultures introduced with 25 μL or 100 μL of the same virus preparations, respectively. ND: No data, experiment not performed; X: No survival. Horizontal asterisks indicate statistical significance by one-way ANOVA test on the percentages of double-positive cells in the selected cultures of the specific split marker vs those in the non-split marker (n.s., non-significant; *$p < 0.05$, **$p < 0.01$, ***$p < 0.001$). Vertical asterisks indicate statistical significance by paired two-sided *t*-test on the percentages of double-positive cells in the selected cultures vs non-selected cultures within each transfection group (n.s., non-significant; *$p < 0.05$, **$p < 0.01$, ***$p < 0.001$)

vectors is to install different fluorescent markers in cells to label different cellular compartments. To explore this application, we cloned in NLS-GFP and LifeAct-mScarlet[9], which label nucleus and F-actin, respectively, by Gateway recombination to conventional full-length (FL) non-split Hygromycin selectable vectors or 2-split Hygromycin Intres vectors. We transduced cells with either sets of plasmids and subjected them to antibiotic selection (Supplementary Fig. 5c). The sample transduced with non-split selectable plasmids contained both singly and doubly labeled cells, while cell transduced with Intres plasmids were all doubly labeled.

**Split mScarlet fluorescent genes for double transgenesis**. To test whether split fluorescent markers can be used for transgene selection, we screened for *Npu*DnaE split points in the mScarlet fluorescent protein (Supplementary Figs. 6 and 7a) and identified four split points allowing for >96% enrichment of double transgenic cells and three other split points enabling >60% enrichment

of double transgenic cells in an mScarlet-gated population, compared to <20% double transgenic cells in non-gated population (Supplementary Fig. 7b, c).

**Three-split Hygromycin Intres for triple transgenesis**. With the split points identified for 2-markertron Intres genes, we set out to engineer higher degree split markers. We tested combinations of splits points to partition a marker gene into three or more markertrons to allow for co-selection of more than two "unlinked" transgenes with one antibiotic (Fig. 4a, b). To identify pairs of split points that would allow such an "Intres chain", we cloned 3-split markertrons into three lentiviral vectors each carrying one of three fluorescent transgenes TagBFP, EGFP, or mCherry, that will allow us to assess effectiveness of selection by flow cytometry (Fig. 4c). Since the Hygromycin resistance gene is the longest and provides the most split points for testing, we focused on engineering 3-split Hygromycin Intres. We tested two 3-split Hygromycin Intres using two intervening *Npu*DnaE

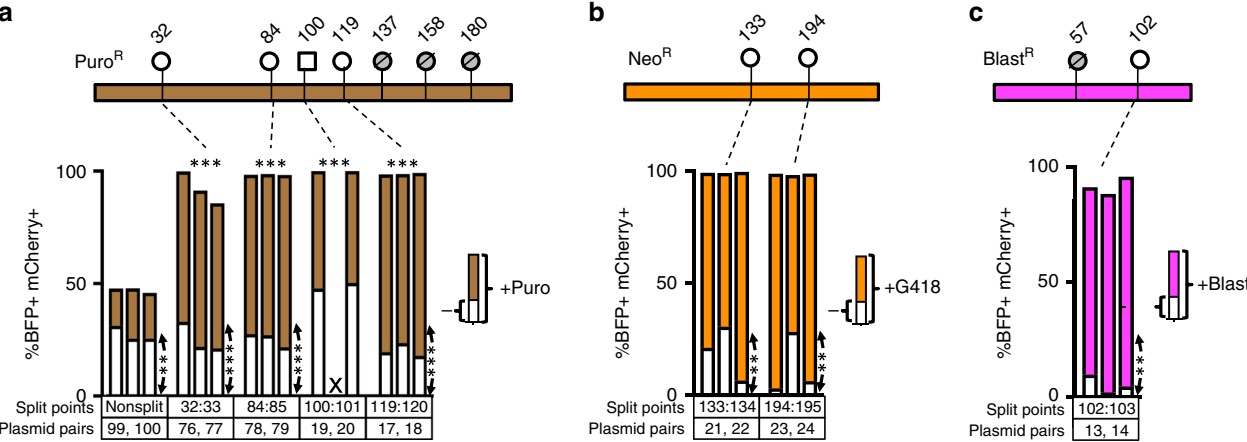

**Fig. 3** Two-Split Puromycin, Neomycin and Blasticidin resistance genes. **a** 2-split Puromycin (Puro) Intres genes. Top schematics show split points tested for Puromycin resistance (Puro[R]) genes while bottom column plot shows percentages of double transgenic cells in the non-selective (white portion) and the selective cultures (total column height = white + brown). Adjacent columns represent results from replicate experiments. X: No survival. **b** 2-split Neomycin/G418 (Neo) Intres genes. Top schematics show split points tested for Neomycin resistance (Neo[R]) genes while the bottom column plot shows percentages of double transgenic cells in the non-selective (white portion) and the selective cultures (total column height = white + orange). Adjacent columns represent results from replicate experiments. **c** 2-split Blasticidin (Blast) Intres genes. Top schematics show split points tested for Blasticidin resistance (Blast[R]) gene while bottom column plot shows percentages of double transgenic cells in the non-selective (white portion) and the selective (total column height = white + cyan) cultures. Results from triplicate experiments are shown as adjacent columns. Horizontal asterisks indicate statistical significance by one-way ANOVA test on the percentages of double-positive cells in the selected cultures of the specific split marker vs those in the non-split marker (n.s., non-significant; *$p < 0.05$, **$p < 0.01$, ***$p < 0.001$). Vertical asterisks indicate statistical significance by paired two-sided $t$-test on the percentages of double-positive cells in the selected cultures vs non-selected cultures within each transfection group (n.s., non-significant; *$p < 0.05$, **$p < 0.01$, ***$p < 0.001$)

inteins (i.e., homogeneous intein), two using *Npu*DnaE for the first intein and *Ssp*DnaB for the second intein, as well as two using *Ssp*DnaB for the first intein and *Npu*DnaE for the second intein (i.e., heterogeneous "orthogonal" inteins) (Fig. 4d). The four heterogeneous-intein 3-split Hygromycin Intres enabled 95–100% triple transgenic selection and the two homogeneous-intein Hygro Intres enabled 74–99% triple transgenic selection in Hygromycin-selected cultures compared to <20% in non-selected cultures. Samples with "leave-one-out" transduction did not yield any viable cells after Hygromycin selection while cells transduced with non-split Hygromycin vectors yielded only 7–17% triple transgenic cells after selection. The observation that 3-split Intres designs using two orthogonal inteins yielded more consistent results than those using the same inteins for the two split points suggest that the use of the same inteins for joining multiple split points may result in artifacts caused by combinatorial splicing that generates "misjoined" fragments. To facilitate the use of 3-split Intres, we created Gateway compatible lentiviral vectors with three of the 3-split Hygromycin Intres (Supplementary Fig. 8a). Three sets of these vectors were each tested by recombining TagBFP (as transgene 1), EGFP (as transgene 2) and mCherry (as transgene 3) into the N-, M-, and C-Intres Gateway destination vectors. Lentiviruses derived from the resultant vectors were used to transduce U2OS cells, which were then split into Hygromycin selective or non-selective media (Supplementary Fig. 8b). Two weeks after selection, cells were analyzed by flow cytometry. All three sets of 3-split Hygromycin Intres plasmids support triple transgenic cell selection of >97% compared to <40% in the non-selected cultures (Supplementary Fig. 8c).

**Application of Intres in CRISPR-Cas-mediated knock-in.** Another potential application of split selectable markers is to facilitate genome engineering and editing via the CRISPR-Cas system[10]. Although gene knockout based on NHEJ-mediated insertions/deletions (indels) occurs at high frequency, precise editing and knock-in based on homology directed repair (HDR)

using exogenous repair templates are inefficient[11]. We tested whether split selectable markers can be used to select for cells with CRISPR-mediated biallelic knock-in at the *AAVS1* locus[12]. We constructed targeting constructs with homology arms flanking the target site, and splice acceptor-2A peptide to trap the markertrons within intron one of the host gene *PPP1R12C*. However, we did not obtain any live cells after CRISPR-Cas knock-in experiments in HEK293T cells using these targeting constructs and two weeks of antibiotic selection. We suspected that the endogenous promoter of the host gene *PPP1R12C* might not drive sufficient expression of markertrons to reconstitute enough antibiotic resistance protein to counter the antibiotic. We thus tested an alternative strategy to express Intres markertrons using the TetO promoter which allows activity to be tuned by doxycycline (dox). To allow comparison of Intres-mediated biallelic selection versus full-length (FL) non-split selectable markers, we implemented several different targeting construct designs. First, we drove expression of a full-length (FL) resistance gene (e.g., Hygro) together with rtTA under a constitutive EF1a promoter and a separate test Intres (e.g., Blast Intres) under a dox-inducible TetO promoter (Supplementary Fig. 9b, Plasmids 109 and 110). This allows comparison of full-length and split selectable markers within the same constructs. To allow valid comparison of full-length versus split markers driven by the same TetO promoter, we constructed two similar plasmids 107 and 108 (cf. Plasmids 109 and 110), wherein the full-length antibiotic resistance gene (Blast) is placed downstream of the TetO promoter. To enable single-cell quantification of biallelic targeting and to demonstrate the feasibility of incorporating two transgenes into two *AAVS1* alleles, we appended EGFP and mScarlet fluorescent genes downstream of the test split or non-split markers via the self-cleaving 2A peptide. Similarly, to test Hygro Intres, we swapped the EF1a and TetO-driven markers so that FL Hygro or Hygro Intres were placed downstream of TetO and FL Blast downstream of EF1a (Supplementary Fig. 9c, d; Plasmids 111–114). We co-transfected pX330-AAVS1 (Plasmid 106)

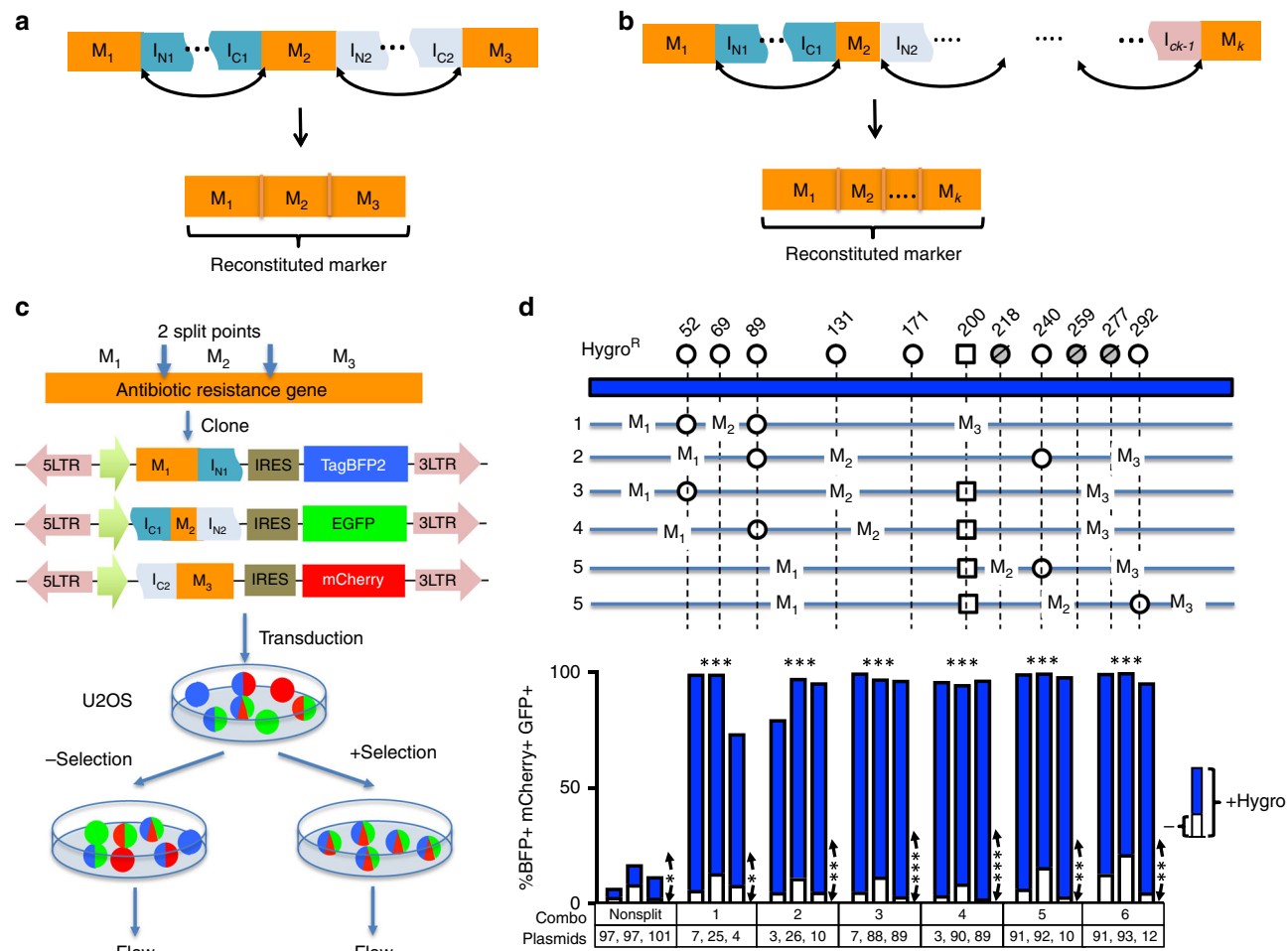

**Fig. 4** Multi-split selectable markers for co-selection of three or more transgenic vectors. **a** Three-split Intres markers. A selectable marker is partitioned into three fragments. The first marker fragment ($M_1$) is fused upstream of the N-terminal fragment of the first split intein ($I_{N1}$). The second fragment ($M_2$) is fused downstream of the C-terminal fragment of the first split intein ($I_{C1}$) and upstream of the N-terminal fragment of the second split intein ($I_{N2}$). The third fragment ($M_3$) is fused downstream of the C-terminal fragment of the second split intein ($I_{C2}$). The intervening split intein catalyzes the joining of the split fragments reconstituting the full selectable marker. **b** A design of a $k$-split selectable marker via an "intein chain" mechanism. The selectable marker is partitioned into $k$ fragments that are reconstituted through protein trans-splicing mediated by intervening split inteins. **c** Split points identified from 2-split selectable markers were used in combination to produce 3-split selectable markers that were cloned into lentiviral vectors with different fluorescent reporters. Cells were then transduced with viruses prepared from these vectors, split into selective or non-selective media. After selection, the cultures were analyzed by flow cytometry. **d** 3-split Hygromycin (Hygro) Intres. Top schematic shows the split points tested for Hygro$^R$, with residue numbers of the last amino acid of the N-terminal fragments indicated above circle or square lollipops, representing *Npu*DnaE and *Ssp*DnaB inteins, respectively. Six designs of 3-split Hygromycin Intres were tested, each indicated with a numbered line with circle or square indicating the two split points used for each design. Column plot below shows the percentages of triple transgenic (BFP + GFP + mCherry + ) cells from the non-selective (white portion) and selective (total column height = white + blue) cultures for the 3-split Hygromycin Intres indicated by the numbers below. Horizontal asterisks indicate statistical significance by one-way ANOVA test on the percentages of triple-positive cells in the selected cultures of the specific split marker vs those in the non-split marker (n.s., non-significant; *$p < 0.05$, **$p < 0.01$, ***$p < 0.001$). Vertical asterisks indicate statistical significance by paired two-sided *t*-test on the percentages of triple-positive cells in the selected cultures vs non-selected cultures within each transfection group (n.s., non-significant; *$p < 0.05$, **$p < 0.01$, ***$p < 0.001$)

containing Cas9 and sgRNA targeting *AAVS1*, and the different pairs of targeting constructs (TC) into HEK293T cells, split into triplicate doxycycline-containing media without antibiotics, with Blasticidin, or with Hygromycin at the subsequent passages. Two weeks after selection, we analyzed the cultures for biallelic targeting by flow cytometric measurement of GFP and RFP fluorescence (Supplementary Fig. 9e). As expected, non-selected cultures harbored a small fraction (<1%) of biallelic knock-in GFP+/RFP+cells (Supplementary Fig. 9e; Selection = None). Selection of antibiotics where corresponding FL antibiotic resistance genes were present on targeting constructs yielded <30% biallelic knock-in cells (Supplementary Fig. 9e; Blast: TC a, c, d;

Hygro: TC a, b, c). In contrast, selection by antibiotics where corresponding Intres are present on the targeting constructs yielded 75% (Supplementary Fig. 9e; Blast Intres: TC b) and 88% (Supplementary Fig. 9e; Hygro Intres: TC d) biallelic knock-in cells. Selection for an additional two weeks allowed split Blast and Hygro TCs to achieve 96.5% and 97.0% biallelic knock-in, respectively (Supplementary Fig. 9f, g). We next tested biallelic engineering in KOLF2-C1 human induced pluripotent stem cells (hiPSCs), which are karyotypically normal with a stable diploid genome[13] (Fig. 5). The full-length non-split Blast targeting constructs (Fig. 5a) and 2-split Blast Intres targeting constructs (Fig. 5b) were tested for selection of biallelically modified clones.

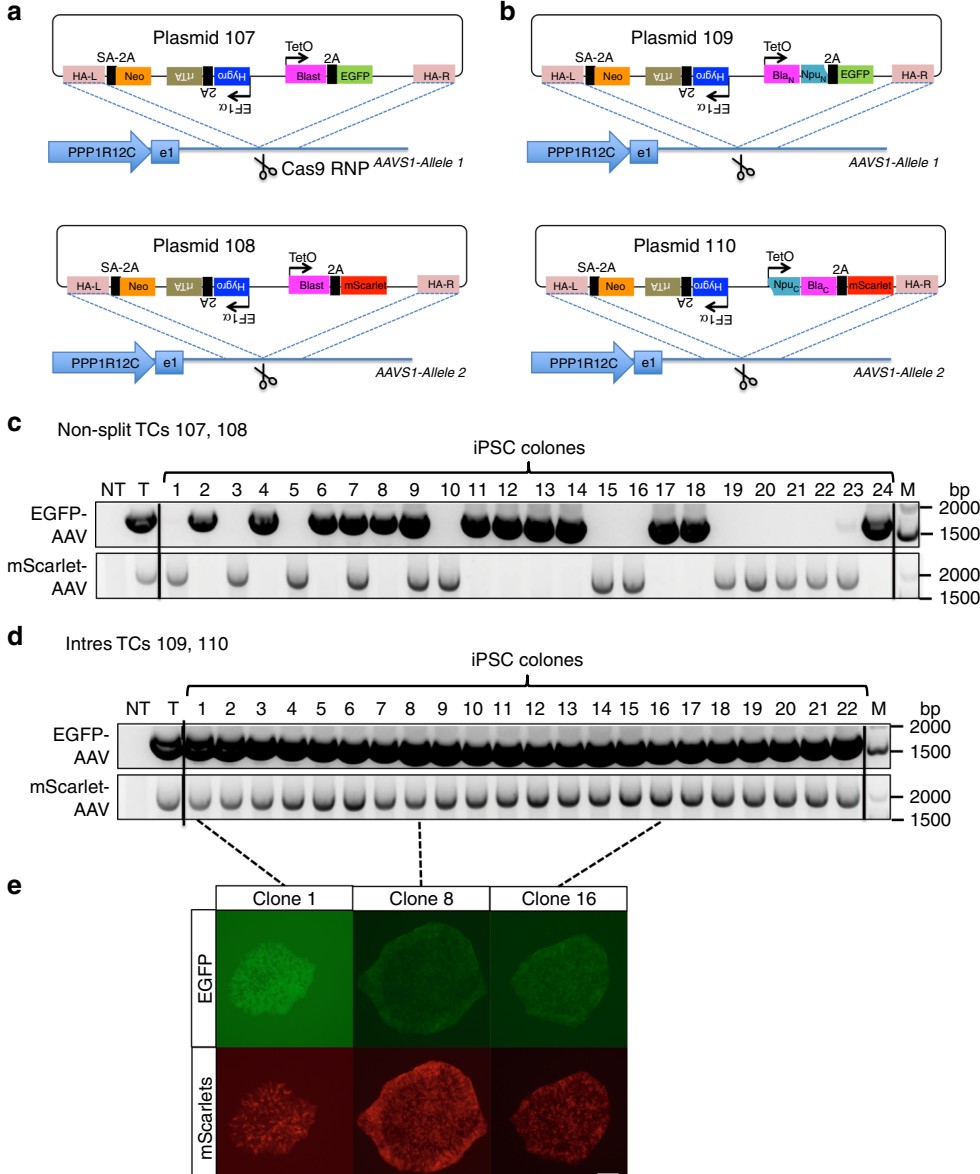

**Fig. 5** Intres for selection of biallelic CRISPR knock-in in human induced pluripotent stem cells. **a** Plasmids 107 and 108 contain AAVS homology arms, and Dox-inducible full-length (FL) Blasticidin (Blast) and EGFP (plasmid 107) or mScarlet (plasmid 108) separated by a self-cleaving 2 A peptide sequence. **b** Plasmids 109 and 110 contain AAVS homology arms, and Dox-inducible N-split Blast markertron and EGFP (plasmid 109) or C-split Blast markertron and mScarlet (plasmid 110) separated by a self-cleaving 2A peptide sequence. Cas9 ribonucleoprotein was formed in vitro by complexing purified Cas9 proteins with synthetic sgRNA targeting AAVS locus, introduced to hiPSCs together with targeting constructs by nucleofection, subsequently subjected to dox-induction and antibiotic selection, and finally cloned by colony picking and analyzed for allelic modification. **c** Genotyping PCR for EGFP- and mScarlet-inserted AAVS alleles of single iPSC clones generated by CRISPR editing using non-split targeting constructs (TCs) 107 and 108. NT: Non-targeted; T: Targeted pool; M: Marker = GeneRuler 1 kb + ladder. **d** Genotyping PCR for EGFP- and mScarlet-inserted AAVS alleles of single iPSC clones generated by CRISPR editing using Intres TCs 109 and 110. NT: Non-targeted; T: Targeted pool; M: Marker = GeneRuler 1 kb + ladder. **e** Representative fluorescent microscopic images of hiPSC colonies from the indicated clones derived from the Intres CRISPR experiments using TCs 109 and 110. Scale bar: 200 μm

Purified Cas9 proteins were complexed with synthetic sgRNA to form Cas9 ribonucleoprotein (RNP) and co-nucleofected with the targeting constructs into KOLF2-C1, followed by dox-induction and antibiotic selection. Surviving colonies were picked into separate wells for establishing single-cell clones. Genotyping PCR revealed that targeting using non-split Blast resistance gene generated only 8% biallelic clones, while targeting using Blasticidin Intres yielded exclusively (100%) biallelically modified clones (Fig. 5c, d), showing both fluorescent signals (Fig. 5e) indicative of the targeting by each targeting construct at the two alleles of AAVS in these hiPSCs.

**Selection of four or more transgenes with Intres**. The utility of Intres may become more apparent in cases where more than three transgenes are to be selected. As we have observed in our 3-split Hygromycin Intres engineering exercise that the use of a set of orthogonal inteins represent a better design for a more robust split marker, we tested four other inteins (gp411, gp418, NrdJ1, IMPDH1)[14] in splitting HygroR or PuroR. We identified additional functional splits of HygroR and PuroR at different positions (Supplementary Figs. 10 and 11). Some of these additional Intres were further adapted to the Gateway cloning system (Supplementary Figs. 12 and 13). To directly observe protein splicing as

well as to confirm these inteins are indeed orthogonal, we conducted western blot analysis of protein trans-splicing between N-markertrons N-terminally tagged with 3xFLAG-epitope and C-markertrons C-terminally tagged with HA-epitope (Supplementary Fig. 14). As expected, while cognate markertrons with matching N- and C-inteins supported reconstitution of the full-length HygroR (lanes 3,5,6), markertrons with unmatched N- and C-inteins did not yield full-length HygroR (lanes 7,8). To introduce and select cells with four or more transgenes, one approach is through sequential transduction/selection of two or more sets of 2-split Intres vectors. By subjecting cells to two rounds of 2-split Intres transduction/selection (Hygro → Puro or Puro → Hygro) with each round carrying two transgenes, we obtained quadruple transgenic cells (Supplementary Fig. 15). These results demonstrated that four transgenes can be sequentially introduced, and that the Intres system is compatible with sequential cell engineering. Another way to introduce four or more transgenes is with higher-degree split Intres markers. By combining the multiple inteins and positions tested for HygroR, we designed and tested 6-split Hygro Intres marker (Supplementary Fig. 16). While cultures transduced with all markertrons yielded viable cells, leave-one-out cultures missing any one of the markertrons did not produce any viable cells after selection. This result demonstrates that up to at least 6 transgenic vectors can be selected simultaneously by one selection scheme using a split selectable marker.

**Proviral copy number analysis**. We validated Intres lentiviral vectors in additional cell lines (HEK293T and HeLa) (Supplementary Fig. 17). To ask whether split markers require a substantially higher copy number than non-split markers to support selection, we conducted proviral copy number analysis in non-selective and selective cultures of cells transduced with non-split HygroR or split Hygro Intres markers (Supplementary Fig. 18) in U2OS, HEK293T and HeLa cells. In general, we observed 1.3–3.1-fold proviral copy numbers in the split marker cultures compared to the non-split cultures. Since the two-split markers require the presence of the two different viral genomes hosting the two markertrons to reconstitute a full resistance protein, it is expected to have ~2-fold equivalence of viral integration to support selection.

## Discussion

In this study, we have engineered split antibiotic resistance and fluorescent protein genes that allow selection for two or more "unlinked" transgenes. By inserting unnatural residues at selectable markers, we showed that additional high-efficiency split points could be utilized, expanding the positions available for engineering. We demonstrated that split selectable markers could be incorporated into lentiviral vectors or gene targeting constructs in CRISPR-Cas9 genome editing experiments for positive selection of cells with double transgenesis or biallelic knock-ins. By combining two splits points, we showed that 3-split markers could be generated to allow higher degree transgenic selection. By conducting sequential transduction/selection with two-split markers, or by combining even more split points we showed the potential to use split selectable markers to select for 4 vectors with two antibiotics or up to 6 vectors with one antibiotic respectively. It is intriguing to anticipate future work to design even higher-degree split selectable markers and to explore the limit of this system for "hyper-engineering" of cells.

## Methods

**Cloning**. To generate a test plasmid for each markertron, we first generated a Gateway donor plasmid containing its ORF and then recombined into lentiviral destination vector with TagBFP2 (Plasmid 94: pLX-DEST-IRES-TagBFP2), EGFP (Plasmid 95: pLX-DEST-IRES-EGFP), or mCherry (Plasmid 96: pLX-DEST-IRES-mCherry) reporters, which were derived from pLX302 (Gift from David Root; Addgene: #25896) by removing Puromycin resistance gene and inserting IRES-fluorescent genes downstream of the Gateway cassette. The markertron-ORF Gateway donor plasmids were generated either by a nested fusion PCR procedure to combine intein with the coding sequence of fragments of the selectable marker followed by insertion into the pCR8-GW-TOPO plasmid by sequence- and ligation-independent cloning (SLIC), or PCR-amplifying the relevant fragment of the selectable marker followed by insertion into "scaffold" plasmids (Plasmids 27~32) containing the intein sequences by SLIC. DNA sequences encoding inteins were codon optimized for *Homo sapiens*, and synthesized as GBlocks (IDT). Selectable marker fragments were amplified from plasmids containing these markers. Plasmids created in this study are listed in Supplementary Table 1 with links to webpages for plasmid sharing and GenBank sequence files.

**HEK293T, U2OS, and HeLa cell cultures**. HEK293T (ATCC® CRL-3216), U2OS (ATCC® HTB-96™), HeLa (ATCC® CCL-2.2™) cells were cultivated in Dulbecco's modified Eagle's medium (DMEM) (Sigma) with 10% fetal bovine serum (FBS) (Lonza), 4% Glutamax (Gibco), 1% Sodium Pyruvate (Gibco), and penicillin-streptomycin (Gibco). Incubator conditions were 37 °C and 5% CO2.

**Virus Production**. A viral packaging mix of pLP1, pLP2, and VSV-G were co-transfected with each lentiviral vector into Lenti-X 293T cells (Clontech/Takara # 632180), seeded the day before in 6-well plates at a concentration of $1.2 \times 10^6$ cells per well, using Lipofectamine 3000. Media was changed 6 h after transfection then incubated overnight. 28 h post transfection, the media supernatant containing virus was filtered using 45 µM PES filters then stored at −80 °C until use.

**Transduction, transfection, flow cytometry, and microscopy**. The day prior to transduction, U2OS, HEK293T, or HeLa cells were seeded into 12-well plates at a density of $1.5 \times 10^5$ cells per well. Prior to transduction, media was changed to media containing 10 µg/mL polybrene, 1 mL per well. In all, 25 µL (or indicated otherwise) of each respective virus (50 µL total for experimental samples with two viruses or 75 µL total for experimental samples with three viruses) was added to each well and incubated overnight. Media was changed 24 h post-transduction. Four days post-transduction, cells were split into duplicate plates. Five days post-transduction, media with antibiotics (130 µg/mL Hygromycin, 2 µg/mL Puromycin, 700 µg/mL G418, or 6 µg/mL Blasticidin) was added to each respective well of one replicate plate (the other remained under no selection). Antibiotics selection continued for 2 weeks before analysis with flow cytometry. For flow cytometry, cells were trypsinized, suspended in media then analyzed on a LSRFortessa X-20 or FACSymphony flow cytometers (BD Bioscience). Fifty thousand events were collected each run. Examplary gating strategy is presented in Supplementary Fig. 19. Microscopy images were taken with the iRiS Digital Cell Imaging System (Logos Biosystems). For transfection for the CRISPR experiment in HEK293T, 600 ng of total plasmids, in equal ratios, were mixed with 100 µL of DMEM and 1.5 µL of attractene (QIAGEN), incubated at RT for 10 min then added to each well and incubated overnight. Media was changed 24 h post-transfection. Two days post transfection, cells were split into duplicate plates with media containing doxycycline (2 µg/mL). Three days post transfection, media with doxycycline and antibiotics was added to each respective well of one replicate plate (the other remained under no selection).

**Human iPSC culture and nucleofection**. Feeder-free KOLF2-C1 (gift from Bill Skarnes, subclone of HipSci HPSI0114i-kolf_2) were maintained on plate coated with Synthemax ll-SC Substrate (Corning) in StemFlex media (Thermo Fisher Scientific). Subculture was carried out every 4–6 days via Accutase (STEMCELL) detachment method. After plating, 1X RevitaCell supplement (Gibco Life Technologies) was added for 1 day to increase cell viability. For Cas9/gRNA ribonucleoprotein (Cas9/gRNA RNPs) and donor plasmids nucleofection, 4D-NucleofectorTM System (Lonza) was used with the P3 Primary Cell 4D-Nucleofector kit (Lonza). Cells were at 60–70% confluency at the time of nucleofection. To assemble Cas9/gRNA RNPs, synthetic single-guide RNA (Synthego) was resuspended in TE buffer (Synthego) at 2 µg/µl, and 8 µl of stock solution was mixed with 20 µg Cas9 protein before nucleofection. For each reaction, $2 \times 10^6$ cells were collected, resuspended in 100 µl complete P3 solution and mixed with pre-assembled Cas9/gRNA RNPs as well as donor plasmids DNA. Doxycycline (Sigma) was added 6 days after nucleofection at 5 µg/mL. Two days after doxycycline was added, 4 µg/mL Blasticidin (Sigma) was applied to select cells with resistance. Surviving single colonies were picked and expanded into Matrigel-coated 24-well plate. If surviving colonies were too large to be manually picked as single colony, cells were replated onto new plate at the density of 2500 cells per 10 cm² plate. Blasticidin treatment continued during single colony expansion in 24-well plates. For genotyping, genomic DNA was extracted using DNeasy Blood & Tissue Kit (QIAGEN), and PCR was performed using the following primers to identify correctly targeted AAVS1 insertions: (i) EGFP-AAVS (fwd: GCCCGACAACC ACTACCTGA, rev: GTGAGTTTGCCCAAGCAGTCA), (ii) mScarlet-AAVS (fwd: CTGAGGTCAAGACCACCTACAAG, rev: GTGAGTTTGCCCAAGCAGTCA). Uncropped gel photos are available within the Source Data file.

**Quantitation of proviral copy number in genomic DNA**. Proviral copy number was measured using Lenti-X Provirus Quantitation Kit (Takara). To perform the analysis, genomic DNA was isolated from transduced cells with NucleoSpin Tissue Genomic DNA Purification (Takara). Serial dilutions of each gDNA sample was subjected to qPCR amplification alongside dilutions of a provirus control template (provided in kit), which was used to generate a standard curve. Since the viral fragments in gDNA and the control template would be amplified with different PCR sensitivities, the provirus copy number was finally calculated based on the standard curve and correlated with a correction factor (provided in manual by Takara).

**Western Blot**. Proteins were separated by 4–20% precast polyacrylamide gel then transferred onto nitrocellulose membrane. After protein transfer, the membranes were incubated in room temperature 5% non-fat milk for 1 h. Membranes were then probed with antibodies against HA-epitope (#3724, 1:1000, Cell Signaling), FLAG-epitope (#F7425, 1:1000, Millipore Sigma), and Vinculin (EPR8185, 1:2000, Abcam). Proteins of interest were detected with HRP-conjugated anti-mouse (9044, 1:20,000, Sigma) and anti-rabbit (0545, 1:20,000, Sigma) and visualized with Clarity™ Western ECL Substrate. Uncropped scans of blots are available in the Source Data file.

**Crystal violet assay**. After virus infection, cells were seeded at 10–15% confluency into 12-well plates in parallel and cultured in hygromycin selection media. Media was changed every 3 days during selection. Crystal violet staining were applied on day0, day3, day5, day7 as well as day14. If cells were greater than 80% (as in the case of sample 1 of Supplementary Fig. 16) confluent on day7, they were split at a 1:20 ratio. For Crystal violet staining, each well was stained with 500 µl 0.1% crystal violet (Sigma) for 10 min at room temperature, then washed gently with 500 µl DPBS for three times before the photographs were taken with an iPhone.

**Reporting summary**. Further information on research design is available in the Nature Research Reporting Summary linked to this article.

## Data availability
The data that support the findings of this study are available from the corresponding author upon reasonable request. Plasmids created in this study are listed in Supplementary Table 1 with links to webpages for plasmid sharing and GenBank sequence files. The source data underlying Fig. 5 and Supplementary Figs. S1b, S14b as well as raw plot numbers are provided as a Source Data file.

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

## Acknowledgements
Research reported in this publication was partially supported by internal funds provided by The Jackson Laboratory as well as grants from the National Human Genome Research Institute (1R01HG009900) and National Cancer Institute (P30CA034196). KOLF2-C1 cells were a gift from Bill Skarnes and were derived from the HipSci consortium. We gratefully acknowledge the contribution of the Flow Cytometry, Cell Engineering Services at The Jackson Laboratory for expert assistance with this publication. Special thanks to our Research Program Development group for assistance with the editing of the manuscript.

## Author contributions
N.J., M.D., J.J.Z., and A.W.C. conceived and designed the study. N.J., M.D., J.J.Z., P.C., and A.W.C. performed the experiments. N.J., M.D., J.J.Z., and A.W.C. analyzed data and wrote the manuscripts.

## Competing interests
A.W.C., N.J., and M.D. filed PCT patent application (WO/2019/075200) for the invention.
