## [Peer Review File · Nature Communications]

Reviewers' Comments:

Reviewer #1:

Remarks to the Author:

The authors describe the engineering of split antibiotic resistance and fluorescent protein genes for selecting two or more "unlinked" transgenes in cultured cells. They successfully demonstrated that split selectable markers could be incorporated into lentiviral vectors or gene targeting constructs in CRISPR/Cas9 genome editing experiments for positive selection of cells with double transgenesis or biallelic knock-ins. Even 3-split markers were possible by combining two split points, thereby allowing higher degree transgenic selection. Overall, the paper is succinct and well written, and the results are interpreted appropriately.

Issues

This reviewer only has one minor criticism. The authors state at the very end that "future development of even higher-degree split selectable markers may enable "hyper-engineering" of cells containing tens of transgenes or targeted knock-ins." It is not immediately clear how one gets from 2 or 3 (as shown here) to tens of knock-ins. And how laborious would this be? This should be more clearly explained or else removed from the text.

Reviewer #2:

Remarks to the Author:

In the manuscript, Nathaniel et al. showed an intein-mediated split marker system for selecting multiple transgene cassettes. They reported to split the marker genes in two or three parts and test their ability in two or three separate integrations and biallelic site modifications. Although splitting selection marker is novel, the split-intein based strategy is a common protein engineering strategy, especially for the virus delivery with limited loading capacity. However, authors did not show a clear benefit from the split selectable marker system compared to the non-split selectable marker system in this manuscript, which makes this manuscript less interesting. A specialized journal would be more suitable for this manuscript.

Major issues

1. Authors only showed the split marker could be used for co-selection.

1.1 The ratio of fluorescence positive cell population is not a right way for measuring the trans-splicing. The authors should show the expression level of the reconstituted marker protein at different split sites compared with the full-length wild-type marker genes.

1.2 Authors should compare the concentration curve of selective drugs using the split system or full-length wild-type system at different virus titer. That means the authors should test whether the low expressed reconstituted marker genes need a higher copy when integration.

1.3 It is interesting that the author adds some amino acid residue to increase the trans-splicing. However, it should be compared by the control group and measured by protein expression level.

2. As the author showed only 2 or 3 integrations in the main text, which could be realized by using different combinations of resistant genes, the author should demonstrate that using the split marker for integrating at least 5 parts together and test the maximum of the number of integrations to show the superiority and Irreplaceability.

3. Based on the point 2 above, the authors should also test using the combinations of different split marker genes.

4. As different cell lines have a different optimal concentration of the selection drug, the authors should demonstrate the multiple integrations at least two different cell lines.

5. The authors did not test the titer of the lentivirus while the integration copy of the virus played an important role in deciding the optimal concentration of the drug. The authors should compare the selection efficiency at the same titer.

6. All the column in the figure did not show the error bar and biological repetition. Has all the experiment only been test in one time?

In theory, the selection will cause survival or death, so the authors should analyze why some column could generate some fluorescence-negative cells.

7. In figure 2, it is interesting to find that the marker gene could be split into three parts. As shown in Figure 2d, only 80% of cells showed triple positive which is different from other columns, so when only some partial pieces connected together, whether it has the resistant function? The author should test it.

8. Similarly, in figure 3, it is not an obvious advantage when using the split marker as the biallelic selection, which could be realized by using two marker genes, the author should test whether they could generate multiple sites biallelic integration simultaneously.

Minor issues,

1. The title and the introduction part of the main text are very concise lack of enough information.

2. Typo in line 66 TagBFP2

3. In the supplementary Figures, it should be attached with supplementary figure legend.

4. The Puro always refer to puromycin (the drug) and PuroR (puro resistant gene) refer to the resistant gene.

5. The blue lines in Supplementary figure 1, 2, 3 and 6 are confusing, the authors should use the space or the other way to show.

6. The bar of the microscope picture should be supplied in the Figure.

Response to reviewers' comments (NCOMMS-18-06957A-Z)

We would like to first express our gratitude to the reviewers' time and effort. We have added substantial data to support and improve the attached manuscript according to the valuable suggestions offered by the reviewers. We would like to highlight changes to the manuscript in response to reviewers' comments point-by-point below:

Reviewers' comments:

Reviewer #1 (Remarks to the Author):

The authors describe the engineering of split antibiotic resistance and fluorescent protein genes for selecting two or more “unlinked” transgenes in cultured cells. They successfully demonstrated that split selectable markers could be incorporated into lentiviral vectors or gene targeting constructs in CRISPR/Cas9 genome editing experiments for positive selection of cells with double transgenesis or biallelic knock-ins. Even 3-split markers were possible by combining two splits points, thereby allowing higher degree transgenic selection. Overall, the paper is succinct and well written, and the results are interpreted appropriately.

Issues

This reviewer only has one minor criticism. The authors state at the very end that “future development of even higher-degree split selectable markers may enable “hyper-engineering” of cells containing tens of transgenes or targeted knock-ins.” It is not immediately clear how one gets from 2 or 3 (as shown here) to tens of knock-ins. And how laborious would this be? This should be more clearly explained or else removed from the text.

We would like to thank reviewer one's encouraging remarks. We agree that we might have overstated the capacity of the split marker system at its current stage. We have thus removed the overstated claim from the discussion and abstract.

Reviewer #2 (Remarks to the Author):

In the manuscript , Nathaniel et al. showed an intein-mediated split marker system for selecting multiple transgene cassettes. They reported to split the marker genes in two or three parts and test their ability in two or three separate integrations and biallelic site modifications. Although splitting selection marker is novel, the split-intein based strategy is a common protein engineering strategy, especially for the virus delivery with limited loading capacity. However, authors did not show a clear benefit from the split selectable marker system compared to the non-split selectable marker system in this manuscript, which makes this manuscript less interesting. A specialized journal would be more suitable for this manuscript.

Major issues

1. Authors only showed the split marker could be used for co-selection.

1.1 The ratio of fluorescence positive cell population is not a right way for measuring the trans-splicing. The authors should show the expression level of the reconstituted marker protein at different split sites compared with the full-length wild-type marker genes.

To directly observe trans-splicing, we have conducted western blot analysis of terminally tagged split markers (**Supplementary Fig. 14**).

1.2 Authors should compare the concentration curve of selective drugs using the split system or full-length wild-type system at different virus titer. That means the authors should test whether the low expressed reconstituted marker genes need a higher copy when integration.

To directly address the copy number requirement, we have conducted proviral copy number analysis of three different cell lines (HEK293T, U2OS, HeLa) transduced with non-split or split markers, that are subsequently cultured in non-selective or selective media (**Supplementary Fig. 19**). We observed 1.3 to 3.1-fold integrations in the split marker-selected cells compared to cells selected via non-split conventional marker. This is a result that we expected since two different viruses are required to reconstitute a full marker for the split systems by design (and also bringing with them two transgenes), while one virus is sufficient to confer resistance in the full-length non-split marker.

1.3 It is interesting that the author adds some amino acid residue to increase the trans-splicing. However, it should be compared by the control group and measured by protein expression level.

In order to allow splicing by NpuDnaE intein, an obligatory “C” residue is added at some positions. We have now shown by western blot that the insertion of C residue is indeed required for splicing (**Supplementary Fig 14, compare lanes 3 and 4**). Another support for the importance of C-insertion is that we did not get resistance/survival without C-insertion for those split marker designs (Data not directly shown).

2. As the author showed only 2 or 3 integrations in the main text, which could be realized by using different combinations of resistant genes, the author should demonstrate that using the split marker for integrating at least 5 parts together and test the maximum of the number of integrations to show the superiority and Irreplaceability.

We have characterized alternative inteins in splitting hygromycin resistance protein. And by combining inteins and split points, we have constructed a 6-split marker (**Supplementary Fig. 17**).

3. Based on the point 2 above, the authors should also test using the combinations of different split marker genes.

We have tested using two-round serial selection using two split markers for different antibiotics, showing that our Intres system is compatible with combination of split markers (**Supplementary Fig. 15**).

4. As different cell lines have a different optimal concentration of the selection drug, the authors should demonstrate the multiple integrations at least two different cell lines.

We have tested split marker lentiviral systems in HEK293T, U2OS, and HeLa cells. In addition, CRISPR-mediated knock-in selection were demonstrated in HEK293T and human iPSCs. In total, we have validated the split marker systems in four different cell types.

5. The authors did not test the titer of the lentivirus while the integration copy of the virus played an important role in deciding the optimal concentration of the drug. The authors should compare the selection efficiency at the same titer.

We use drug concentration that we have optimized using the full-length antibiotic resistance gene in the different cell lines and are the “standard” in our lab. These concentrations are also within the range published in the literature. As mentioned above, we have also looked directly at proviral integration copy number to compare the requirement for both full-length non-split marker vs our split markers (**Supplementary Fig. 19**).

6. All the column in the figure did not show the error bar and biological repetition. Has all the experiment only been test in one time?

In theory, the selection will cause survival or death, so the authors should analyze why some column could generate some fluorescence-negative cells.

We have included replicate data in most figures.

7. In figure 2, it is interesting to find that the marker gene could be split into three parts. As shown in Figure 2d, only 80% of cells showed triple positive which is different from other columns, so when only some partial pieces connected together, whether it has the resistant function? The author should test it.

In our leave-one-out experiments, we did not see cell survival if any one of the pieces were left out. However, we do not know why there are escapees in the selection. One postulation we have is that the 3-split markers using the same inteins for the two split points may have complications arising from unintended splicing products. We observed more robust selection in 3-split markers using different inteins for the two split points. We have added description of some possible explanations to the text.

8. Similarly, in figure 3, it is not an obvious advantage when using the split marker as the biallelic selection, which could be realized by using two marker genes, the author should test whether they could generate multiple sites biallelic integration simultaneously.

We appreciate suggestion to further show the system for integrating multiple sites in the context of CRISPR gene editing. However, we argue that it is out of the scope of the current study and should be continued in a new project. We hope that the current manuscript has conveyed

sufficient support of the split marker system using lentiviral delivery to introduce 2 or 3, and up to the potential of 6 transgenes.

Minor issues,

Thank for pointing these minor issues. We have corrected/changed most of them.

1. The title and the introduction part of the main text are very concise lack of enough information.

We appreciate the reviewers' recognition of the conciseness of the introduction. We hope to give the reader a concise introduction with references so that they can refer to when more details are needed.

2. Typo in line 66 TagBFP2

Thank you for pointing out the typo. We have corrected it.

3. In the supplementary Figures, it should be attached with supplementary figure legend.

Done.

4. The Puro always refer to puromycin (the drug) and PuroR (puro resistant gene) refer to the resistant gene.

Corrected.

5. The blue lines in Supplementary figure 1, 2, 3 and 6 are confusing, the authors should use the space or the other way to show.

Thank you for the suggestions. We would however like to respectfully request to stick with our style.

6. The bar of the microscope picture should be supplied in the Figure.

Scalebars are added.

Reviewers' Comments:

Reviewer #2:

Remarks to the Author:

I still have concerns regarding the detailed description of the new selection marker systems. Each pair of split markers (including PuroR, hygRO, BlaR, NeoR etc.) should be described in details about the decreased expression level of full length protein in different cell lines and dose curve change of suitable selection drug. These results will be useful for users who use the split marker in different cell lines and conditions. And the split point with highest expression of full-length protein for each selection marker genes should be clearly indicated in the manuscript.

In addition, the quality of data is not very solid, and the figure is perplexing. Do the 2-4 columns mean different repeats, e.g. in Fig. 2 and why some columns are labeled as X? Why the difference between different columns within one group is very diverse e.g. in Figure 2? In almost all the supplementary Data, the number of repeats seems to be no more than 2. Three repeats are highly commended for these data. The number of biologic repeats should be consistent (all ≥ 3 e.g. in Figure 2 some data with twice repeat, some data with 2-4 repeats and some data with 3 repeats), and the error bar of experimental results and statistics analysis should be clearly indicated in all figures. In addition, the presentation of the data e.g. in Fig. 2 (all labeled with plasmid number rather than the description) is very confusing and hard to read.

For a method paper, an interesting application is necessary to demonstrate the advantage of using split marker system over combinations of multiple selective markers. I appreciate the supplementary Figure 17, but the only microscopy data is not convincing. Authors should demonstrate whether it can work like the results in the main figure with three biologic repeats and ideally show the quantity data.

Response to reviewers' comments (NCOMMS-18-06957C)

We would like to first express our gratitude to the reviewers' time and effort. We have added substantial data to support and improve the attached manuscript according to the valuable suggestions offered by the reviewer #2. We would like to highlight changes to the manuscript in response to the reviewer's comments point-by-point below:

Reviewer #2 (Remarks to the Author):

I still have concerns regarding the detailed description of the new selection marker systems. Each pair of split markers (including PuroR, hygR, BlaR, NeoR etc.) should be described in details about the decreased expression level of full length protein in different cell lines and dose curve change of suitable selection drug. These results will be useful for users who use the split marker in different cell lines and conditions. And the split point with highest expression of full-length protein for each selection marker genes should be clearly indicated in the manuscript.

Thank you for the suggestion from the reviewer to look at protein splicing directly. We have now conducted and included western blot analysis of 5 split hygromycin designs (Suppl Fig 1b) and found that the less robust resistance conferred by one of the split points (131) may be correlated to its lower splicing efficiency. This is a very useful addition to the paper. Thanks! However, as we now discuss in the text, splicing efficiency might only be one of the factor affecting resistance activity and that protein folding before/after splicing might also affect the function of the resistance protein. Thus, the functional test - resistance/selection efficiency followed by flow cytometry readout of transgenic expression – should be the ultimate proof for the usability of the markers and is our primary method for evaluating these split markers. As we have shown in our viral copy number analysis, we saw 1.3~3.1x viral integration in the split systems compared to the non-split. This is expected since we need to integrate two different vectors to achieve resistance while full-length markers require only 1 vector to confer resistance. Therefore, functionally (conferring resistance), even if the splicing reaction is not 100% efficient, the split markers are not “disadvantaged” compared to the non-split markers.

As for the concentration of drug to use for selection for different cell lines, one must find the optimized concentration by performing kill curve analysis on non-transduced cells for each cell type [1], i.e., the drug concentration used for each cell line is not a function of the marker, but rather the cell type and the drug. For each of the three cell lines (HEK293T, HeLa and U2OS) tested, drug concentration was identified for each cell line for each antibiotic by performing kill curve experiments as described in [1] on non-transduced (“wild-type:WT”) cells when we first started using these cells and has been a standard in our lab for years. These concentrations have been effective in any cases. In this study, we use the same drug concentration for selecting cells transduced with the non-split or split markers, i.e., we do not need a lower concentration of drug to select for the split marker system to compensate for incomplete splicing-reconstitution of markers. With our examples in four cell lines, I would predict that for other cell lines used by other users, full-length and split marker system can use the same drug concentration, which is determined per cell line using kill-curve experiments in WT non-transduced cells.

[1] <https://www.sigmaaldrich.com/technical-documents/articles/biology/antibiotic-kill-curve.html>

In addition, the quality of data is not very solid, and the figure is perplexing. Do the 2-4 columns mean different repeats, e.g. in Fig. 2 and why some columns are labeled as X? Why the difference between different columns within one group is very diverse e.g. in Figure 2? In almost all the supplementary Data, the number of repeats seems to be no more than 2. Three repeats are highly commended for these data. The number of biologic repeats should be consistent (all ≥ 3 e.g. in Figure 2 some data with twice repeat, some data with 2-4 repeats and some data with 3 repeats), and the error bar of experimental results and statistics analysis should be clearly indicated in all figures. In addition, the presentation of the data e.g. in Fig. 2 (all labeled with plasmid number rather than the description) is very confusing and hard to read.

The neighboring columns are biological replicate experiments, as we described in the figure legend. We also described in the figure legend the meaning of "X" being no cell survival. To make it clearer, we have included keys in the figure legend to help the readers understand the figures.

To increase the robustness of our study, we have conducted more repeat experiments and included statistical analysis where appropriate. Furthermore, for reproducibility and transparency, we display all replicate data points in the figures, not merely showing summary statistics. All plot data are also included in raw data table supplement (Source Data file) so that interested readers can perform other analysis on the data. The robustness of the split markers are also supported by the following: For many split marker designs (e.g., Hygro-Npu89), in addition to testing a particular vector with biological replicates (new viral preps, new transduction, different experiment dates), we have also tested different marker-transgene architecture, e.g., marker-IRES-transgene (e.g., Fig 2b column group 2) and transgene-IRES-marker (e.g., Supplementary fig 5b, Hygro), with their respective replicates, demonstrating their effectiveness. In addition, we have tested our lentiviral vectors in three different cell lines (HEK293T, U2OS, HeLa) and also for different engineering approach (CRISPR in hiPSC).

We provide plasmid numbers in figures referencing those in the supplementary table 1 with links to Addgene or other plasmid information page, so that people can conveniently obtain the exact plasmids and reproduce our findings or apply our system in their projects. We also have a schematic diagram on top of the column plots to show the correspondence of each column and the split point in the antibiotic resistance protein. With also the supplementary figures 1~4 showing the amino acid sequences of the resistance proteins and the split points cross referencing plasmid numbers.

For a method paper, an interesting application is necessary to demonstrate the advantage of using split marker system over combinations of multiple selective markers. I appreciate the supplementary Figure 17, but the only microscopy data is not convincing. Authors should demonstrate whether it can work like the results in the main figure with three biologic repeats and ideally show the quantity data.

As we have opened in the introduction, there are limited choices of commonly used antibiotics in mammalian cell cultures (Hygro, Puro, Neo, Blast). Our 6-split markers allow for "one-shot" selection of 6 vectors, which cannot be done even with the combination of all four different resistance genes, so this is in our opinion a big leap. And there is no way to compare these two (6-split vs 4-markers). Unfortunately, our flow cytometry cannot differentiate >4 fluorescent proteins. We may think of another application to apply six transgenic selection, but it will be a completely new story that will potentially take another year or two to complete. We hope to quickly communicate our results so that users can already use this powerful technology in their

work, and we hope to spark interests for other engineers to optimize or innovate new split marker systems. We have repeated the selection experiments three times now and included crystal violet staining experiment for the course of 14 days during one experiment (Suppl Fig 16). This result clearly demonstrate survival only in samples receiving all six plasmids.